# Adaptation and psychometric properties of Psychological Skills Inventory for Sport (PSIS-R5) in Latvian athletes: Insights and implications for practice

Katrina Volgemute[1*◉], Gundega Ulme[2◉], Viktorija Perepjolkina[2◉], Renars Licis[1‡], Agita Abele[1‡], Rodrigo Lavins[1‡], Alīna Klonova[1,3], Anna Tihija[1], Everts Grants[1]

1 Latvian Academy of Sport Education, Riga Stradiņš University, Riga, Latvia, 2 Department of Health Psychology and Pedagogy, Riga Stradiņš University, Riga, Latvia, 3 Department of Neuroscience, Biomedicine, and Movement, University of Verona, Verona, Italy

◉ These authors contributed equally to this work.
‡ RL, AA, and RL also contributed equally to this work.
* katrina.volgemute@rsu.lv

## Abstract

### Background and objective

Psychological skills are critical for high achievement and elite sports performance. However, there remains a lack of valid, reliable, and accessible instruments adapted to modern athletes. This highlights significant psychometric challenges in the existing tools currently available. While the Psychological Skills Inventory for Sport (PSIS-R5) is a commonly used tool in sports psychology worldwide, it has demonstrated various psychometric issues over time. A previous adaptation to the Latvian sports environment highlighted specific challenges in the Latvian context. It is important to note that these issues are not unique to the Latvian adaptation. A comprehensive review is necessary to address these broader psychometric concerns and ensure the instrument's suitability for contemporary athletes, including those in Latvia. Moreover, the adaptation process may offer insights and methodological guidance for similar efforts in other cultural contexts. The aims of this research were twofold: first, to review and adapt the PSIS-R5 for use in Latvia (PSIS-R5-L), ensuring its validity and reliability; and second, to assess the applicability of the PSIS-R5-L in elite sports settings by examining relationships between athletes' achievement levels and psychological skills.

### Methods

A total of 444 Latvian athletes aged between 18–43 (M = 21.32; SD = 6.52) completed the PSIS-R5 inventory to assess their psychological skills. The PSIS-R5 was re-translated into Latvian language (PSIS-R5-L) using forward-backward translation.

**Data availability statement:** The dataset supporting the conclusions of this article is available at the Riga Stradiņš University Dataverse repository: Volgemute K, Ulme G, Perepjolkina V, Licis R, Abele A, Lavins R. Psychological Skills Inventory for Sport (PSIS-R5) for Latvian Athletes [Dataset]. Riga, Latvia: Riga Stradins University Dataverse; 2025. https://doi.org/10.48510/FK2/ZUFA2Q. For long-term access or institutional inquiries, data requests may also be directed to the RSU Research Department at research@rsu.lv.

**Funding:** The author(s) declare financial support was received for the research, authorship, and/or publication of this article. This research is funded under the Grant No. RSU/LSPA-PA-2024/1-0010 of the project No. 5.2.1.1.i.0/2/24/I/CFLA/005 "RSU Internal and RSU with LASE External Consolidation" (funded by the European Union Recovery and Resilience Facility and the budget of the Republic of Latvia). The funders had no role in study design, data collection and analysis, decision to publish, or preparation of the manuscript.

**Competing interests:** The authors declare that they have no known competing financial interests or personal relationships that could have appeared to influence the work reported in this paper.

Additionally, five sports psychologists participated in the study, providing evaluations of elite athletes' psychological skills using the PSIS-R5 inventory.

## Results

Series of Principal Components Analyses (PCA) were conducted to explore and validate the structure of the Latvian revisited adaptation of the PSIS-R5-L. As a result, a stable four-factor structure was obtained, consisting of 17 items with satisfactory fit indices (CFI = 0.968, RMSEA = 0.045 [90% CI: 0.034, 0.057], SRMR = 0.026). The findings indicate that the Latvian adapted PSIS-R5-L has appropriate psychometric properties, confirming its suitability for use in the Latvian sports environment. Comparisons between different athlete groups (elite, pre-elite, and amateur) revealed several correlations and statistically significant ($p < 0.05$) differences in psychological skills.

## Conclusions

The PSIS-R5-L is a psychometrically robust and practically valuable tool for assessing athletes' psychological skills in Latvia. It offers coaches, sports psychologists, and practitioners an evidence-based instrument to evaluate and enhance mental skills across different athlete populations, supporting psychological preparation programs in both elite and amateur sports settings.

## Introduction

Sports are becoming increasingly demanding, requiring athletes to be well prepared not only physically, technically, and tactically but also psychologically to achieve success [1,2]. The study of psychological factors plays a crucial role in athletic performance, highlighting the importance of understanding athletes' psychological skills. This understanding enables the optimization of training processes and the identification of individual needs, ultimately contributing to the long-term improvement of athletic performance [3,4]. The psychological preparation of athletes focuses on enhancing psychological skills and developing techniques to help them perform to their full potential under pressure. Therefore, it is crucial to reliably and validly identify and measure psychological skill indicators using appropriate assessment tools tailored to the demands of the modern sports environment. In sports psychology one of the most pressing challenges in the modern sports environment is the lack of high quality, appropriately adapted and validated instruments for specific sports populations. As noted by Cid et al. this issue often stems from the use of measurement tools that fail to hold on to culturally or regionally approved criteria or lack a solid conceptual foundation [5]. Addressing these shortcomings requires focused efforts to identify methodological gaps and refine approaches to the adaptation or standardization of instruments, ensuring their continued relevance in sports psychology. To accurately assess psychological skills in sports, along with their interrelationships to athletic performance, it is essential to use measurement tools that are valid, reliable, and precise [6].

Self-assessment instruments demand careful and thoughtful adaptation or standardization across diverse cultural contexts. The process of adapting effective measurement tools extends beyond simple translation [5,7]. It necessitates a comprehensive evaluation of psychometric properties to ensure accuracy and relevance. A common issue today is the continued reliance on outdated sport psychology measurement instruments that, while widely recognized in the sports community, may no longer resonate with modern athletes or reflect current trends in sports psychology. Previous studies have emphasized the need for updated validation process to ensure psychological measurement tool accuracy to reflect current psychological constructs [8]. Reviews and updates of these tools are essential to align them with the evolving sports environment and to ensure their terms and constructs are comprehensible and meaningful to today's athletes. This can help maintain the relevance of psychological assessments that ultimately can contribute to better research and practical outcomes in the field of sport.

Studies confirm that psychological skills are a set of trainable psychological abilities that help athletes to enhance their performance [9,10]. Existing literature suggests that the Psychological Skills Inventory for Sport (PSIS-R5) stands out as the most useful instrument in evaluating athletes' psychological skills [11]. The PSIS-R5, developed by Mahoney et al. is an instrument designed to measure six psychological skill aspects of athletes in a sports environment, and it demonstrates good psychometric properties and is primarily used to assess athletes' psychological skills through various cognitive strategies [12]. This psychological skills measurement tool consists of six subscales, including Anxiety Control, Concentration, Self-Confidence, Visualization, Motivation, and Team Emphasis.

The PSIS-R5 is a well-established tool in sports science research, frequently used to assess athletes' psychological skills, examine their connections with other performance variables, and predict athletic outcomes [13]. The inventory was initially designed to differentiate between elite, pre-elite, and amateur athletes based on six key psychological factors: Mental Preparation, Motivation, Concentration, Self-Confidence, Team Emphasis, and Anxiety Control. Research following the development of the original inventory revealed that elite athletes exhibit higher motivation, lower levels of negative anxiety, greater use of visualization techniques, and a stronger focus on individual performance compared to pre-elite or amateur athletes [12].

More recently, an adaptation of the PSIS-R5 was conducted with Indonesian national athletes. This adaptation resulted in a framework of five factors: Self-Management, which integrates confidence, self-control, and concentration, Motivation, Psychological Readiness, Thoughts of Failure or Defeat, and Team Management [11]. These results reflect cultural and contextual differences while maintaining the core aim of assessing psychological skills in sports. Another version of the PSIS-R5 was developed by Milavic et al. to measure psychological characteristics, specifically focusing on young athletes, as this demographic is more likely to reveal differences between talented and less talented athletes [13]. According to the authors, this revised instrument retained the original six-factor structure of the PSIS-R5 but reduced the number of items. The selection process emphasized content validity, with experts identifying three representative items for each factor, resulting in a total of 18 items. This streamlined structure was statistically reliable and content-valid, making it well-suited for the moder sports environment for young athletes while remaining easy to comprehend.

The PSIS-R5 was previously adapted to the Latvian sports environment by Fernate [14]. During this adaptation process, the author identified several ambiguities in the psychological skills measurement instructions, test forms, and item content. The inventory was not fully adapted but rather adjusted to fit the sports environment in Latvia. These shortcomings were confirmed by the issues in psychometric properties of inventory, which highlighted the need for a thorough review and refinement. It is also worth noting that this first adaptation took place over 15 years ago, further underlining the necessity of updating the instrument to align with current sport practices. The increasing global demand for standardized, valid psychological assessment tools in sport highlights the importance of such adaptations. Although this version was created for Latvian athletes, its design process and findings offer insights relevant for broader international application, particularly in countries with emerging sport psychology infrastructures.

## The role of psychological skills in sport

Psychological skills play a crucial role in athletes' success and achievement. For this reason, they are widely studied in sports science and sports psychology, particularly in high-performance sports. Research on psychological skills often aims to achieve different aims, such as examining the effects of psychological training interventions on athletic performance [15,16]. Psychological skills training for optimal sports performance has garnered significant attention from athletes, coaches, and experts, leading to numerous studies on the topic. Park and Jeon, in their bibliometric analysis of psychological skills, grouped various themes into four key clusters. These include (1) training aimed at reducing stress, enhancing mental toughness, and improving coping mechanisms, (2) managing anxiety, motivation, self-confidence, and self-efficacy, (3) fostering flow states and mindfulness (4) and regulating emotions [17]. These themes highlight the multifaceted approaches to psychological skills training, all converging on the goal of optimizing athletic performance through strategies such as stress management, anxiety control, and emotional regulation. Their findings confirm that psychological skills training focuses on optimizing performance through various strategies, including stress management, anxiety control, and coping mechanisms, among others. These core aspects are also addressed by the PSIS-R5. Mahoney's PSIS-R5 inventory for assessing psychological skills is designed with a focus on elite or high-performance sports, making it highly relevant in sports science for understanding what distinguishes successful athletes from less successful ones.

Another significant area of research involves examining differences in psychological skills across different groups of athletes and their levels of achievement. For instance, Mitic et al. explored whether psychological skills differ between elite and non-elite athletes. Their findings revealed that elite-level athletes exhibit higher self-efficacy (belief in their abilities), greater emotional openness, and a stronger future time perspective. They also demonstrate lower levels of negative past time perspective and higher emotional competence compared to non-elite athletes [18]. Research indicates that psychological skills and their related indicators provide valuable insights into athletic performance and achievements. Moreover, these skills can be used as predictors of success in sports [19]. Consequently, understanding and developing psychological skills remain integral to enhancing both individual and team performance in competitive sports as well as in elite, pre-elite and amateur level sports.

It is important to note that, in the Latvian context, elite athletes are also studied in various aspects of psychological skills and techniques. For example, Eikena and Iancheva conducted a study specifically focused on Latvian elite athletes [20]. While this topic and related research are undoubtedly highly relevant, it must be acknowledged that there is a significant need for new and modern instruments tailored to the modern sports environment. Such tools are essential for accurately assessing psychological skill levels in athletes, particularly at the elite sports level.

## Study aims and hypothesis

Keeping in mind previously mentioned, the aim of this study was twofold: first, acknowledging that previous research has identified psychometric issues in the PSIS-R5 inventory and recognizing earlier attempts to adapt it to the Latvian sports context, this study aimed to review and adapt the PSIS-R5 for use in Latvia (PSIS-R5-L), ensuring the inventory's validity and reliability. Second, to assess the applicability of the PSIS-R5-L in elite sports settings by examining the relationships between athletes' achievement (elite, pre-elite and armature) levels and psychological skills. While the focus of the adaptation is on Latvian athletes, the development process and resulting tool may also serve as a model for researchers and practitioners in other countries with similar needs, contributing to the advancement of cross-cultural assessment of psychological skills in sport. It was hypothesized that there will be significant differences in psychological skills across different levels of achievement (elite, pre-elite, and amateur) as assessed by the adapted PSIS-R5-L inventory.

The PSIS-R5 inventory is widely used for both practical applications and research purposes. However, in Latvian sports psychology, there is a significant gap due to a lack of culturally relevant and modern instruments for accurately assessing psychological factors without the risk of outdated findings. Updating and adapting this inventory for today's athletes seeks to provide a valid tool that addresses known limitations with the current version in practical settings. This

adaptation intends to resolve previously observed challenges and ensure the inventory's relevance for contemporary Latvian athletes. In the sports environment, there are several factors that highlights the need for culturally adapted psychological assessment tools. Latvia has a relatively small population, resulting in a limited sample of elite athletes who often combine professional sport with academic or work responsibilities. Access to psychological support remains limited compared to larger countries, which places greater emphasis on athletes' psychological preparation. Athletes frequently face challenges such as financial constraints and a need for greater resilience and adaptability. These contextual factors underscore the importance of having psychometric tools that are sensitive to the specific psychological demands faced by athletes. This ensures that assessments are accurate, culturally appropriate, and practically meaningful within this unique sporting environment.

## Methods and materials

### Participants

This study involved a sample of 444 athletes aged 17–43 years, with a mean age of M = 21.49 (SD = 5.68). Of the 444 athletes, 271 (61%) were male, 170 (38.3%) were female, and 3 (0.7%) identified as other. The average experience in competitive sports was M = 7.67 years (SD = 5.4), with an average weekly training load of M = 7.78 hours (SD = 5.21).

The athletes represented a total of 58 different sports, including hockey (n = 73), basketball (n = 52), floorball (n = 36), volleyball (n = 35), football (n = 33), handball (n = 25), athletics (n = 18), fitness (n = 16), luge (n = 15), swimming (n = 13), figure skating (n = 13), orienteering (n = 11), sport dance (n = 9), rugby (n = 8), judo (n = 8), gymnastics (n = 8), beach volleyball (n = 7), curling (n = 6), table tennis (n = 5), boxing (n = 3), tennis (n = 3), and others (n = 46).

For the assessment of test-retest reliability, a subsample of 40 athletes was included. These athletes were proportionally drawn from the full sample to ensure representation across elite (n = 12), pre-elite (n = 14), and amateur (n = 14) levels. The research sample also included Latvian psychologists (n = 5) who actively work in the field of sports psychology. Out of the 10 sports psychologists registered with the Latvian Sports Psychology Association, only those engaged in active practice were selected for the study.

### Measurements

This study utilized the Psychological Skills Inventory for Sports (PSIS-R5), a self-report measure designed to assess athletes' psychological skills. The original English version of the PSIS-R5 includes 45 items distributed across six scales: Anxiety control, Concentration, Confidence, Mental preparation, Motivation, and Team emphasis [12]. All across the scale, respondents are required to rate each item on a five-point Likert scale ranging from 1 (completely disagree) to 5 (completely agree). Sample items include "When I make an error in my performance, I become very anxious" (Anxiety control scale) and "When I mentally practice I "see" myself performing (well) just like I was watching a videotape" (Mental preparation scale).

The PSIS-R5 is widely applied in both sports practice and sports psychology research to evaluate athletes' psychological skills and has been adapted for use in various countries, resulting in several alternative versions based on the original construct [13]. In Latvia, the inventory was previously adapted by researcher Fernate, who noted issues with the psychometric properties of the Latvian version, indicating a need for further in-depth analysis to address these challenges [14].

### Translation of the PSIS-R5

The PSIS-R5 inventory was previously translated into Latvian in 2008. While this version provided an initial adaptation, it became evident that certain terms, such as "heat waves", were not well understood by younger respondents. This indicates that the translation did not fully align with contemporary Latvian language usage or cultural context. To address this, the PSIS-R5 original version was translated directly from the original source and carefully reviewed the 2008 version

to ensure all prior insights were considered while improving accessibility and relevance for modern audiences. To successfully adapt the PSIS-R5 instrument for the Latvian sports environment, the process involved several steps (Fig 1). The first task was translating and culturally adapting the inventory into Latvian language. This was done by selecting two independent translators, both of whom are native Latvian speakers. Each translator produced a separate version of the translation.

Next, both Latvian translations were subjected to a thorough review by experts to ensure semantic, conceptual, and content accuracy, confirming that the translations were consistent with the original meaning of the items. After these revisions, an independent translator performed a reverse translation of the Latvian version back into English. The reverse translation was then compared with the original English version to ensure that the translated items retained their original meaning and intent. Once the accuracy and similarity were confirmed, a face validity of PSIS-R5 Latvian version was checked. Three athletes completed the Latvian version of the inventory and provided feedback on the meaning of each item. Based on the analysis of the athletes' responses, face validity was confirmed. The final version of the PSIS-R5 in Latvian (PSIS-R5-L) was then developed, completing the adaptation process.

## Procedure, design and ethics

In the next steps of the research, the developed PSIS-R5-L was administered to athletes. Data collection was conducted using a web-based questionnaire administered via Microsoft Forms, as well as through paper format inventories that were completed by hand. The target population consisted of athletes from various sports and competitive levels. The

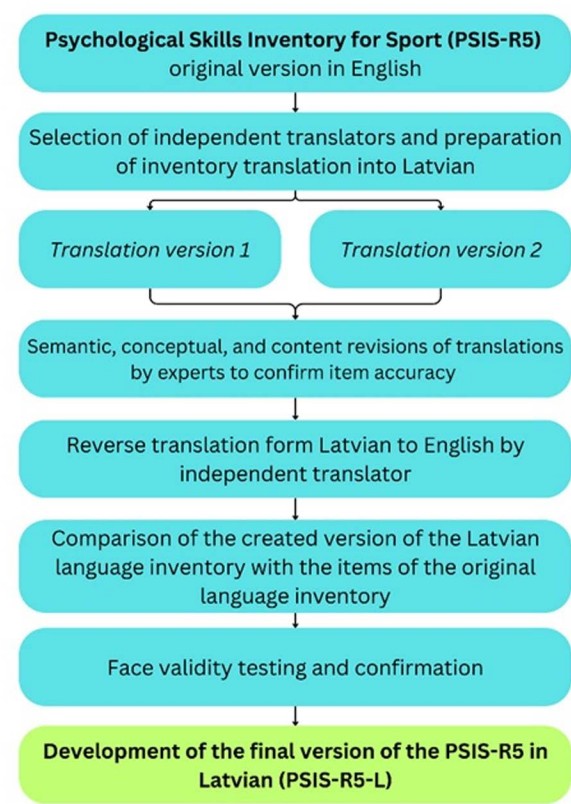

**Fig 1. PSIS-R5-L translation and validation processes.**

participants recruitment and data collection period lasted approximately five months, beginning on 30 April 2024 and concluding on 30 September 2024. The inventory was distributed directly to the targeted athletes. In addition to completing the inventory and evaluating each of its items, participants were asked to provide demographic information, including age, gender, city of residence, sport type, weekly training hours, level of achievement, and training experience in their specific sport.

Participation in the study was voluntary and anonymous, with participants being informed that their data would be used solely within the framework of this research. This study was conducted after receiving approval from the Ethics Committee of the Latvian Academy of Sport Education (Protocol No. 8, Statement No. 1, April 19, 2024) and in accordance with the ethical principles outlined in the Declaration of Helsinki. Prior to completing the questionnaire, participants provided written informed consent, ensuring they were aware of the study's purpose and the intended use of their data. No minors participated in this study, and all participants were competent adults who voluntarily consented to partake. Confidentiality was strictly maintained, with all data anonymized and securely stored to protect participant privacy. Data was encrypted and stored on secure servers, accessible only to authorized research team members. The study adhered to data protection regulations as outlined in registered data management plan with the ARGOS (OpenAIRE) system, ensuring full compliance with legal and ethical standards. Participants were informed of their right to withdraw from the study at any point without penalty. In such cases, any identifiable data would be immediately deleted to uphold their privacy rights.

After data collection, they were analyzed using appropriate mathematical statistical analysis following the example of creating the original version [12].

## Statical analysis

During the adaptation of the PSIS-R5-L inventory, 467 responses were initially collected, of which 23 were deemed invalid due to incomplete responses, leaving 444 valid responses for further analysis [21]. The data collected was analyzed using the Statistical Package for the Social Sciences (IBM SPSS v.28.0.0) and JASP 0.18.3.0. Statistical methods were selected based on recommendations in the scientific literature as well as the guidelines established by the original inventory developers to ensure a reliable and valid adaptation for the Latvian sports context. To assess data normality, the Kolmogorov-Smirnov test was applied, confirming that the assumptions for parametric statistics were met. Principal Component analysis was conducted using the varimax rotation method, aligning with the original development approach. Model fit was evaluated using Comparative Fit Index (CFI), and Root Mean Square Error of Approximation (RMSEA), where a CFI above 0.95 and an RMSEA below 0.06 indicated an adequate fit [22]. Following factor analysis, data modeling was refined through Standardized Parameter Estimates in a Structural Equation Modeling (SEM) framework to further validate the factor structure. Due to violations of normality, bootstrapping was employed within the SEM framework. Bootstrapping resampling technique was used to obtain robust standard errors, confidence intervals, and average path estimates, ensuring more reliable parameter estimation and model evaluation. The internal consistency of the PSIS-R5-L factors was assessed using Cronbach's Alpha. Values exceeding 0.6 are typically considered acceptable, while coefficients between 0.50 and 0.80 are regarded as indicating moderate reliability [23]. In addition to Cronbach's alpha, Composite Reliability (CR) and Average Variance Extracted (AVE) were computed to further evaluate the internal consistency and convergent validity of the factors. CR values greater than 0.70 and AVE values greater than 0.50 are considered indicative of good reliability and convergent validity. However, according to Fornell and Larcker, AVE values slightly below 0.50 can still be accepted if CR exceeds 0.70 [24]. The temporal reliability of PSIS-R5-L was evaluated using the Intraclass Coefficient (ICC), with reliability deemed acceptable at an ICC value of 0.7 or higher, as recommended by Hausenblas et al. [25].

In line with a secondary study objective, athletes were grouped by achievement level (elite, pre-elite, and amateur). The independent samples T-tests were used to assess statistically significant differences in psychological skills across various athletes' achievement level groups. Finally, bivariate Pearson's correlation tests examined relationships between athletes' demographic factors, training load, experience, sport type (individual vs. team), and the factors in the Latvian PSIS-R5-L.

Sample size adequacy was confirmed using G*Power 3.1.9.6 software, ensuring a 5% α-error level and 80% power (equivalent to a 20% β-error rate).

## Results

### Factor analysis and structural model: exploring of PSIS-R5-L structure for Latvian sports environment

This study aimed to adapt the PSIS-R5 inventory for the Latvian sports environment, creating a reliable and valid tool for assessing athletes' psychological skills. Earlier attempts to adapt the PSIS-R5 inventory to the Latvian context revealed that its factor structure did not align with the original version. Fernate [14] highlighted several issues that warranted further investigation into the inventory's psychometric properties.

To examine the factor structure of the PSIS-R5-L, a series of Principal Component Analyses (PCA) with varimax rotation was conducted. In the initial stage, all 45 items of the translated PSIS-R5-L were subjected to PCA to assess whether the planned six-factor structure could be obtained, as in the original inventory. The analysis was performed twice: first, determining the number of factors based on eigenvalues greater than 1, and second, based on parallel analysis. When the number of factors was determined based on eigenvalues greater than 1, a 12-factor solution was obtained. Thirteen items loaded onto the first factor, while four unique items loaded onto each of the second, third, and fourth factors. Three unique items loaded onto each of the fifth, sixth, and seventh factors. Two unique items loaded onto the tenth and twelfth factors. The eighth, ninth, and eleventh factors each had only one item. Additionally, four items had factor loadings below 0.40 on all factors. The analysis was repeated using parallel analysis. In this method, factors are selected when their eigenvalues exceed the corresponding parallel average random eigenvalues. This approach yielded a six-factor solution: 21 items loaded onto the first factor, four unique items loaded onto each of the second, fourth, and fifth factors, five items loaded onto the third factor, and two items loaded onto the sixth factor. Additionally, five items had factor loadings below 0.40 on all factors. The solution appeared promising and closer to the original structure. Given that parallel analysis is considered a superior method for selecting factors compared to traditional methods based on eigenvalues greater than 1 [26], the analysis was continued using this approach to determine the number of factors.

In the next steps, five consecutive PCAs were performed sequentially. In the first step, five items with low loadings were removed, resulting in a four-factor solution where seven items exhibited low loadings on all factors. In the next step, these seven items were removed, yielding the same four-factor solution but with one item showing low loadings across all factors. This item was then removed, producing a stable four-factor solution. This solution consisted of 20 items in the first factor, five items (including one with cross-loading) in the second factor, three items in the third factor, and six items (including one with cross-loading) in the fourth factor. In the final step, the 13 items with lower factor weights from the first factor were excluded, leaving the seven items with the highest factor weights (excluding item 32, as its content was very similar to other questions). During the adaptation process, a substantial item reduction was performed. Out of the original 45 items in the PSIS-R5, only 17 were retained in the final PSIS-R5-L structure. This decision was based on psychometric criteria (e.g., factor loadings below 0.40, cross-loadings) and the cultural appropriateness of items for the Latvian sports environment. A total of 28 items were excluded. Table 1 summarizes the original items retained, their original factor allocation, and changes where applicable. Notably, several items originally part of the Anxiety Control and Concentration factors (Q20, Q21, and Q38) loaded onto the newly defined Self-Confidence factor, and the Mental Preparation factor was renamed to Visualization. Thus, while maintaining the core psychological constructs of the original instrument, the PSIS-R5-L presents a modernized and psychometrically robust structure. A full mapping of the retained items, their original factor assignments, and any changes in the adapted PSIS-R5-L version is provided in S2 File. The factor loadings and descriptive statistics of the retained 17 items, along with the resulting four-factor structure, are presented in Table 1.

The four-factor model of the PSIS-R5-L, based on the 17 retained items, was evaluated for its fit with the respondents' sample data and its suitability for factor analysis. The KMO = .824 and statistically significant Bartlett's Test of Sphericity (p < 0.001) approves the adequacy of sample and that the correlation matrix of the variables in dataset diverges

**Table 1. Item content, factor that the item belongs to, standardized factor loading (λ), uniqueness and internal consistency.**

| Item | Content | M | SD | λ | Uniqueness |
|---|---|---|---|---|---|
| F1: Self-Confidence (α = 0.87; CR = 90; AVE = 0.56) | | | | | |
| Q30 | When I begin to perform poorly, my confidence drops very quickly | 2.84 | 1.23 | 0.82 | 0.32 |
| Q38 | When I make an error in my performance, I become very anxious | 2.92 | 1.18 | 0.80 | 0.35 |
| Q18 | It doesn't take much to shake my self-confidence | 2.83 | 1.24 | 0.78 | 0.37 |
| Q36 | My self-confidence jumps all over the place | 3.13 | 1.22 | 0.76 | 0.42 |
| Q21 | When I make a mistake, I have trouble forgetting it and concentrating on my ongoing performance | 2.86 | 1.16 | 0.74 | 0.45 |
| Q28 | I have frequent doubts about my athletic ability | 2.76 | 1.25 | 0.70 | 0.48 |
| Q20 | I am often panic struck during those last few moments before I begin my performance | 2.26 | 1.25 | 0.63 | 0.56 |
| F2: Motivation (α = 0.72; CR = 0.82; AVE = 0.53) | | | | | |
| Q39 | Right now the most important thing in my life is to do well in my sport | 3.17 | 1.38 | 0.83 | 0.31 |
| Q42 | My sport is my whole life | 3.25 | 1.29 | 0.77 | 0.40 |
| Q1 | I am very motivated to do well in my sport | 4.30 | 0.9 | 0.67 | 0.45 |
| Q12 | Winning is very important to me | 4,00 | 1.06 | 0.63 | 0.51 |
| F3: Team Emphasis (α = 0.71; CR = 0.82; AVE = 0.60) | | | | | |
| Q27 | I enjoy working with teammates | 4.38 | .83 | 0.82 | 0.30 |
| Q10 | I get along very well with other members of a team | 4.31 | .82 | 0.80 | 0.35 |
| Q31 | I think team spirit is very important | 4.56 | .83 | 0.70 | 0.47 |
| F4: Visualization (α = 0.50; CR = 0.74; AVE = 0.48) | | | | | |
| Q13 | I often "rehearse" my performance in my head just before I perform | 3.77 | 1.14 | 0.77 | 0.38 |
| Q33 | When I mentally practice I "see" myself performing (well) just like I was watching a videotape | 3.62 | 1.13 | 0.69 | 0.48 |
| Q35 | When I am preparing to perform I try to imagine what it would feel like in my muscles | 2.59 | 1.26 | 0.62 | 0.59 |

*Notes*: Applied rotation method is Varimax. Principal Component Analysis based on Principal Components, KMO 0.824. Number of factors based on Parallel Analysis based on FA, Factoring method: Maxiimum likelihood. M = Arithmetic Mean; SD = Standard Deviation; α = Cronbach's Alpha; CR = Composite Reliability; AVE = Average Variance Extracted.

significantly from the identity matrix. The initial eigenvalue for the first factor of the four-factor solution, which accounted for 23.4% of the variance after rotation, was 4.19. The eigenvalue for the second factor (13.2% of the variance after rotation) was 2.68, the eigenvalue of the third factor (12.0% of the variance after rotation) was 1.62, and the eigenvalue of the fourth factor (9.1% of the variance after rotation) was 1.33. Obtained four-factor structure cumulatively explain 57.7% of the total variance. These results suggest that a four-factor structure is suitable for the data (see Table 2).

Based on the results of the Principal Component Analysis the four-factor structure identified through factor analysis demonstrated good stability. This analysis identified four distinct factors of PSIS-R5-L: Self-Confidence, Motivation, Team Emphasis, and Visualization (see Table 3).

**Table 2. Total variance explained of PSIS-R5-L.**

| Extraction Sums of Squared Loadings | | | |
|---|---|---|---|
| Factor | Total | % of Variance | Cumulative % |
| F1 | 4.189 | 23.4% | 23.4% |
| F2 | 2.678 | 13.2% | 36.6% |
| F3 | 1.620 | 12% | 48.6% |
| F4 | 1.329 | 9.1% | 57.7% |

*Notes*: Extraction Method: Principal Component Analysis

**Table 3. Model additional fit indices for the factor analysis of the PSIS-R5-L.**

| RMSEA | RMSEA 90% confidence | SRMR | TLI | CFI | BIC |
|---|---|---|---|---|---|
| 0.045 | 0.034–0.057 | 0.026 | 0.0941 | 0.968 | −309.651 |

*Notes:* RMSEA: The Root Mean Square Error of Approximation; SRMR: Standardized Root Mean Square Residual; TLI: Tucker-Lewis Index; CFI: The Comparative Fit Index; BIC: Bayesian Information Criterion.

The first factor of the four-factor structure was defined as F1: Self-Confidence scale, which includes items Q30, Q38, Q18, Q36, Q21, Q28 and Q20. Items Q30, Q18, Q36, and Q28, with factor loadings ranging from 0.7 to 0.82, aligned with the original version's Self-Confidence scale. However, item Q38 and Q20, which factor loading of 0.80 and 0.63 on this scale, was originally part of the Anxiety Control scale. Analyzing the significance of Q38 items content, "When I make an error in my performance, I become very anxious" it can be inferred that this item also closely relates to athletes' self-confidence and their ability to maintain confidence under pressure. This item highlights how performance errors can trigger anxiety, potentially undermining self-assurance, which is crucial for optimal performance. Similarly, the content of Q20 item "I am often panic struck during those last few moments before I begin my performance" indicates that pre-competition anxiety can affect an athlete's confidence. The connection between this item and self-confidence is essential, as feelings of panic can impede an athlete's ability to execute their skills effectively. Both items serve to emphasize the interplay between anxiety and self-confidence, illustrating how anxiety can detract from an athlete's belief in their capabilities. Another item that rotated into the Self-Confidence scale was Q21, which was originally part of the Concentration scale. The content of item Q21, "When I make a mistake, I have trouble forgetting it and concentrating on my ongoing performance," also implies through close analysis that its content relies on self-confidence structures. This item underscores the impact of mistakes on an athlete's ability to focus, suggesting that a lack of self-confidence can hinder concentration and performance. When athletes struggle to move past errors, it can create a cycle of self-doubt and anxiety, further affecting their confidence levels. Thus, Q21 reinforces the idea that concentration and self-confidence are intricately linked, as effective performance relies not only on skill but also on the athlete's psychological state and ability to recover from setbacks. It is possible that Latvian athletes, within their cultural context, interpret these items differently, associating them more closely with their levels of self-confidence. Such cultural nuances can influence how athletes perceive and respond to items measuring psychological constructs like self-confidence and anxiety.

The second factor was defined as F2: Motivation, which includes items Q39, Q42, Q1 and Q12, with factor loadings ranging from 0.63 to 0.83. These four items also constituted the Motivation scale in the original version of PSIS-R5.

The third factor was defined as F3: Team Emphasis, consisting of Q27, Q10 and Q31 items, with factor loadings ranging from 0.70 to 0.83. All items align with the corresponding scale in the original English-language version.

In the original PSIS-R5 inventory, the fourth scale is referred to as Mental Rehearsal. After evaluating the scale's content and considering terminology for psychological skills in the Latvian language, the decision was made to rename this fourth factor as F4: Visualization, comprising items: Q13, Q33 and Q35 with factor loading ranging from 0.62 to 0.77. All items correspond to the Mental Rehearsal scale in the original version.

In order to assess model fit, Comparative Fit Index (CFI), Root Mean Square Error of Approximation (RMSEA), and Standardized Root Mean Square Residual (SRMR) were analyzed. According to Schermelleh-Engel et al. models with a CFI ≥ 0.95, RMSEA ≤ 0.08, and SRMR ≤ 0.10 are considered to have an "acceptable" fit, while models with a CFI ≥ 0.97, RMSEA ≤ 0.05, and SRMR ≤ 0.05 are considered to have a "good" fit [27]. The results obtained indicate that the model's fit for the factor analysis can be evaluated as acceptable (see Table 3).

Following factor analysis, Structural Equation Modeling (SEM) was conducted to validate the factor structure, assess the relationships between latent constructs, and evaluate the overall model fit of the Latvian revisited adaptation of the

PSIS-R5-L. Given the non-normality of the data, bootstrapping was employed (replication N = 5000) to ensure robust parameter estimates and accurate significance testing.

Fig 2 presents the estimates of the four-factor structure of the PSIS-R5-L version. The results from the factor analysis indicated strong and acceptable standardized factor loadings (λ). The bootstrap with p-values less than 0.001 and 95% confidence intervals (C.I.) indicated the stability of the factor loadings. All PSIS-R5-L items had significant factor loadings on their hypothesized latent factors.

### Reliability analysis

The reliability analysis, conducted using Cronbach's alpha coefficients, demonstrated high internal consistency across most scales of the PSIS-R5-L in the developed version, with one exception. The coefficients ranged from 0.87 to 0.50. The Self-Confidence, Motivation, and Team Emphasis scales exceeded the commonly accepted cutoff point of α = 0.7 for social psychology studies. The Visualization scale, while lower at α = 0.5, falls within the moderate reliability range (0.50–0.80) as suggested by Salvucci et al., although caution is recommended when interpreting this scale [22]. Furthermore, Composite Reliability (CR) and Average Variance Extracted (AVE) were calculated to evaluate internal consistency and convergent validity. All scales showed satisfactory CR values: Self-Confidence (CR = 0.90), Motivation (CR = 0.82), Team Emphasis (CR = 0.82), and Visualization (CR = 0.74), each exceeding the minimum of 0.70. The AVE values indicated acceptable convergent validity for Self-Confidence (AVE = 0.56), Motivation (AVE = 0.53), and Team Emphasis (AVE = 0.60). Although the AVE value for Visualization (AVE = 0.48) was slightly below the ideal cutoff of 0.50, it is considered acceptable in combination with the CR value above 0.70, according to Fornell and Larcker's criteria [24]. This indicates acceptable to strong inner reliability and consistency in the measurement of constructs.

Compared to the previous Latvian adaptation of the PSIS-R5 [14], the current PSIS-R5-L demonstrated improved psychometric properties. In the earlier adaptation, only two subscales (Self-Confidence, Motivation) showed acceptable internal consistency (α = 0.80 and α = 0.65, respectively), while the other subscales had Cronbach's alpha coefficients ranging from 0.46 to 0.52, and the Visualization subscale showed a negative alpha coefficient (α = −0.24), indicating internal consistency issues. In contrast, the PSIS-R5-L demonstrated Cronbach's alpha coefficients ranging from 0.50 to 0.87, with three out of four subscales exceeding α = 0.70. Additionally, the PSIS-R5-L factor structure demonstrated an acceptable model fit (CFI = 0.968, RMSEA = 0.045, SRMR = 0.026), supporting stronger convergent and discriminant validity compared to previous adaptations. These results suggest that the item reduction and restructuring enhanced the psychometric quality of the adapted inventory. The PSIS-R5-L represents a psychometrically enhanced and culturally relevant adaptation of the original instrument for Latvian athletes.

After the PSIS-R5-L structure was established, test-retest temporal reliability analyses were conducted to assess the consistency of its scales over a 21-day period on a sub-sample of 40 athletes. The test-retest reliability of the PSIS-R5-L was evaluated using ICC for four of its scales: Confidence (ICC = 0.893), Motivation (ICC = 0.833), Team emphasis (ICC = 0.881), and Visualization (ICC = 0.715). All scales demonstrated ICC values above 0.7, indicating good temporal stability.

### Identification and comparison of Latvian language version of PSIS-R5-L inventory results among athletes across different achievement levels

The second aim of this study was to assess the differences in psychological skills among athletes of varying achievement levels in sport by using the PSIS-R5-L inventory. For further analysis, the athletes were categorized into three groups: elite, pre-elite, and amateur. This classification was guided by the original PSIS-R5 inventory by Mahoney et al., where similar criteria were used to stratify athletes, and adapted to fit the specific characteristics of the Latvian sports environment following the recommendations of sports experts [12].

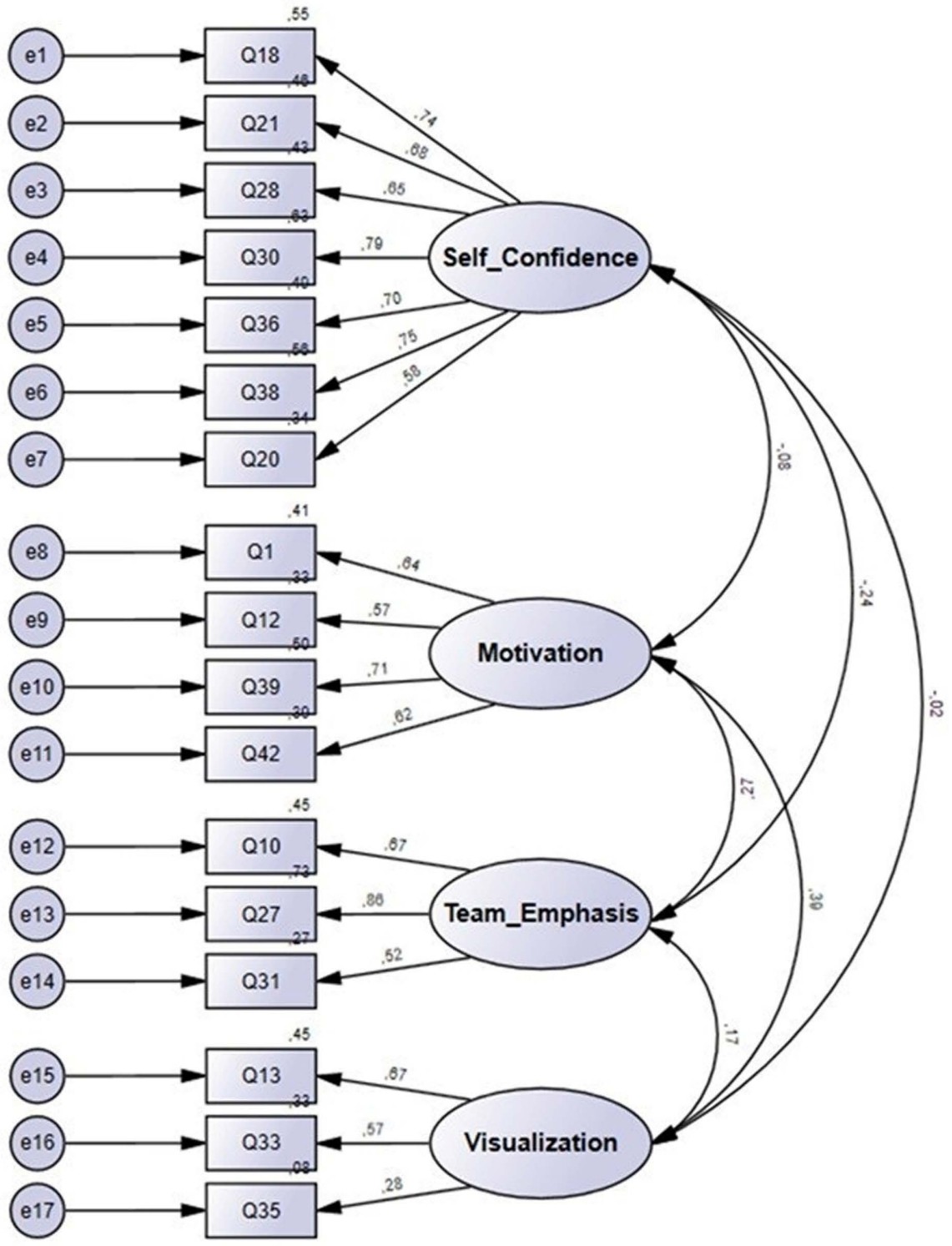

**Fig 2. Bootstrap standardized coefficients.** *Notes*: Correlation between factors in the four-factor PSIS-R5-L model with 17 items. All estimates p<0.001; the numbers within squares represent the original numbers of the items in the PSIS-R5-L scales (as shown in Table 2).

**Table 4. Descriptive statistics and differences among different athlete level groups in the Latvian version of PSIS-R5-L factors.**

| Factor | | Descriptive scores | | | | | | Independent Samples T-Test | | |
|---|---|---|---|---|---|---|---|---|---|---|
| | | Elite (n = 25) | | Pre-elite (n = 219) | | Amateur (n = 200) | | Elite vs Pre-elite | Pre-elite vs Amateur | Elite vs Amateur |
| | | M | SD | M | SD | M | SD | | | |
| F1: | Self-Confidence | 3.09 | 0.81 | 2.68 | 0.86 | 2.90 | 0.97 | 2.31* | −2.54* | 0.94 |
| F2: | Motivation | 3.98 | 0.74 | 3.78 | 0.84 | 3.54 | 0.88 | 1.16 | −2.76* | 2.38* |
| F3: | Team Emphasis | 4.52 | 0.47 | 4.45 | 0.67 | 4.37 | 0.67 | 0.53 | 1.21 | 1.10 |
| F4: | Visualization | 3.65 | 0.85 | 3.32 | 0.88 | 3.29 | 0.76 | 1.82 | 0.27 | 2.18* |

*Notes:* M = Arithmetic Mean; SD = Standard Deviation;

*Correlation is significant at the 0.05 level (2-tailed). Maximum points for each scale: 5.

For the elite group (n = 25; 5.6% of the sample), athletes were selected based on criteria that emphasized a high-quality, professional training regimen. Specifically, elite athletes participated in at least eight training sessions or trained for no less than 12 hours per week. Furthermore, these athletes were required to meet the following performance criteria: being a prize-winner in the National Championship (higher league) for adults, achieving a top placement in regional leagues (such as the Baltic League), or competing in international events like the European Championship (junior or adult levels), World Cup, World Championship (junior or adult levels), or the Olympic Games. Additionally, an athlete's experience in their respective sport needed to be at least five years.

The pre-elite (n = 219; 49.3% of the sample) group was selected with criteria focused on a more moderate training load, requiring no less than five sessions or 7.5 hours per week. Their sports performance needed to include participation in National Championships (higher leagues) at any age, or in regional leagues or university-level competitions (Universiade).

Finally, the amateur (n = 200; 45% of the sample) group was defined by a minimum training load, with athletes engaging in at least two training sessions or three hours of training weekly. Their competitive experience was limited to participation in medium or low-level competitions at any age.

By employing these criteria, athletes were effectively categorized according to their levels of achievement. This classification facilitated a comparison of psychological skills across different levels of athletes, allowing for an evaluation of whether the developed Latvian version of the PSIS-R5-L inventory could distinguish between these groups based on its psychometric indicators. The study's sample size was determined using G*Power. The sample size was estimated using the analysis of variance test (F test) for three groups with a 5% α-error level and an 80% power level (which corresponds to a β-error rate of 20%). The results indicated that a total of 231 athletes were required, with 77 athletes per group. However, given the stringent criteria established for identifying elite athletes to ensure evaluation of the highest-level athletes, the size of this group was smaller than would typically be acceptable. In contrast, the other athlete groups significantly exceeded the required size, resulting in a total sample size well above the initial requirement. Table 4 presents the results of the t-test analysis used to assess the psychological skills indicators among athletes on the adapted Latvian language version of the PSIS-R5-L inventory scales.

The obtained data reveal that there are significant differences in self-confidence between elite and pre-elite athletes (t = 2.313, p < 0.05), suggesting that elite athletes demonstrate a higher level of self-confidence. Additionally, statistically significant differences in self-confidence were found between pre-elite and amateur athletes (t = 2.760, p < 0.05), indicating that pre-elite athletes exhibit lower self-confidence than amateur athletes. No statistically significant differences were found in self-confidence between elite and amateur athletes (p > 0.05). An analysis of the average scores on the self-confidence scale shows that the elite group has the highest mean (M = 3.09; SD = 0.81), followed by the amateur group (M = 2.90; SD = 0.97), with the pre-elite group scoring the lowest (M = 2.68; SD = 0.86). The motivation scale, based on average scores, shows a descending order from the elite (M = 3.98; SD = 0.74) to the pre-elite (M = 3.78; SD = 0.84), with

amateurs (M = 3.54; SD = 0.88) group scoring the lowest. Statistically significant differences in motivation were observed between pre-elite and amateur athletes (t = −2.525, p < 0.05) and between elite and amateur athletes (t = 2.381, p < 0.05). These results suggest that athletes with higher levels of achievement exhibit higher motivation in sports. On the team emphasis scale, no statistically significant differences were detected among athletes' achievement groups, indicating comparable team orientation across all groups. However, based on average scores, elite (M = 4.52; SD = 0.47) athletes scored highest on this scale, followed by pre-elite (M = 4.45; SD = 0.67) and amateur (M = 4.37; SD = 0.67) athletes with slightly lower scores. For the visualization scale, statistically significant differences were found only between the elite and amateur groups (t = 2.184, p < 0.05), suggesting that elite athletes possess stronger visualization skills than amateurs. The average scores on this scale reflect a similar pattern, with elite (M = 3.65; SD = 0.85) athletes scoring the highest, followed by pre-elite (M = 3.32; SD = 0.88), and amateurs (M = 3.29; SD = 0.76) scoring the lowest. These findings align with previous studies, supporting that the higher achievement levels are associated with stronger psychological skills.

## Elite athlete self-assessment compared to sports psychologists' evaluations

To gain a deeper understanding of psychological skill differences among athletes at varying levels of achievement, particularly within elite sports, five leading sports psychologists in Latvia were invited to complete the Latvian version of the PSIS-R5-L-inventory. Their input was sought to provide professional perspectives on what constitutes an elite or top-level athlete in professional sports. Given their extensive experience with athletes across different achievement levels, these psychologists offer valuable insights into whether the inventory results differ meaningfully among athletes at various stages. Additionally, their expertise enhances the interpretation of these findings within the context of elite performance.

Table 5 presents the results of sports psychologists' evaluations of what describes an elite level athlete, including mean scores, standard deviations, and t-test results comparing these evaluations with those provided by elite athletes themselves. This analysis follows a similar approach used by Mahoney et al. in comparing perspectives from experts and athletes. The results reveal that, for most PSIS-R5-L inventory scales there are no statistically significant differences between the evaluations of sports psychologists and elite athletes. However, the self-confidence scale stands out as an exception, with a significant difference detected (elite athletes: M = 3.08; SD = 0.81, sports psychologists: M = 1.6; SD = 0.62, t = −3.892, p < 0.05). This finding suggests an interesting conclusion that sports psychologists perceive elite athletes' self-confidence to be considerably lower than the athletes themselves report. This difference in perceived self-confidence may indicate that sports psychologists, who are trained to observe athletes' internal struggles and vulnerabilities, recognize underlying self-doubt that elite athletes might not readily disclose or even acknowledge. It is possible that elite athletes report higher self-confidence because their environment often expects and emphasizes confidence as a key trait for gaining success in sport. This social expectation may lead athletes to internalize and project greater self-confidence, aligning with the resilience needed to succeed in high-stakes, competitive conditions.

**Table 5. Descriptive statistics and differences between elite level achievement group and sport psychologist evaluation in PSIS-R5-L factors.**

| Factor | | Descriptive scores of sport psychologist evaluation (n = 5) | | Independent Samples T-Test |
|---|---|---|---|---|
| | | M | SD | |
| F1: | Self-Confidence | 1.60 | 0.62 | −3.89* |
| F2: | Motivation | 4.30 | 0.65 | 1.03 |
| F3: | Team Emphasis | 4.47 | 0.45 | 0.99 |
| F4: | Visualization | 4.20 | 0.76 | 0.85 |

*Notes:* M = Arithmetic Mean; SD = Standard Deviation;

*Correlation is significant at the 0.05 level (2-tailed).

The other scales of PSIS-R-L inventory: motivation (elite athletes: M = 3.98; SD = 0.7, sports psychologists: M = 4.3; SD = 0.65, t = 1.034, p > 0.05), team emphasis (elite athletes: M = 4.52; SD = 0.47, sports psychologists: M = 4.47; SD = 0.45, t = 0.992, p > 0.05), and visualization (elite athletes: M = 3.65; SD = 0.85, sports psychologists: M = 4.2; SD = 0.76, t = 0.854, p > 0.05), did not show statistically significant differences. This alignment between elite athletes' self-assessments and the evaluations from sports psychologists lends support to the inventories reliability in measuring psychological skills especially in the context of elite sport.

**Interrelationships between athletes' demographics, achievement levels and psychological skills**

The Pearson correlation results displayed in Table 6 exhibited a significant relationship among factors included in the analysis. The magnitude of the relationship in Pearson correlation results was considered either weak (r > 0.1), moderate (r > 0.3), or strong (r > 0.5), in accordance with the standard suggestions provided by Cohen [28].

The Pearson correlation results in Table 6 show significant relationships between the factors analyzed. The analysis of Pearson correlation reveals a correlation between athletes' achievement levels and motivation, indicating that athletes with higher achievement levels exhibit greater motivation (r = −0.155, p < 0.01). This finding aligns with the t-test results. No significant correlations were observed with other PSIS-R5-L inventory scales regarding to athletes' achievement levels. Interestingly, a negative correlation was found between team emphasis and self-confidence (r = −0.198, p < 0.01), suggesting that athletes who prioritize teamwork tend to report lower self-confidence, and the other way around. Conversely, team emphasis and motivation were positively correlated (r = 0.212, p < 0.01), indicating that athletes with a strong team orientation also demonstrate higher motivation, highlighting the importance of an athlete's social environment for motivation in sports. Motivation and visualization also showed a positive correlation (r = 0.225, p < 0.01), suggesting that visualization skills, which are integral to mental toughness and psychological preparation of athletes, are closely related to motivation. Visualization is known to be a challenging yet crucial psychological skill that enhances competitive performance. The Pearson correlation analysis also shows a negative association between age and both self-confidence (r = −0.118, p < 0.05) and motivation (r = −0.248, p < 0.01), indicating that younger athletes tend to report higher self-confidence and motivation. Similar gender-based correlations were observed. Female athletes showed higher self-confidence (r = 0.174, p < 0.05), while male athletes displayed higher motivation (r = −0.138, p < 0.01). No significant correlations were found between training load and any psychological skill indicators (p > 0.05). However, athlete experience was negatively

**Table 6. Pearson correlation matrix between athletes' achievement levels, PSIS-R5-L factors, age, gender, training load, experience and sport type.**

| | Factor | 1 | 2 | 3 | 4 | 5 | 6 | 7 | 8 | 9 | 10 |
|---|---|---|---|---|---|---|---|---|---|---|---|
| 1 | Level | 1 | | | | | | | | | |
| 2 | Self-Confidence | .054 | 1 | | | | | | | | |
| 3 | Motivation | −.155** | −.063 | 1 | | | | | | | |
| 4 | Team Emphasis | −.070 | −.198** | .212** | 1 | | | | | | |
| 5 | Visualization | −.066 | −.001 | .225** | .101* | 1 | | | | | |
| 6 | Age | .026 | −.118* | −.249** | −.048 | −.046 | 1 | | | | |
| 7 | Gender | −.010 | .174* | −.138** | −.074 | .015 | −.001 | 1 | | | |
| 8 | Training Load | −.270** | −.116* | .090 | .030 | .000 | .238** | −.148** | 1 | | |
| 9 | Experience | −0.99* | −.163** | .072 | .086 | .102* | .203** | −.113* | −.327** | 1 | |
| 10 | Sport type (individual vs team) | .021 | −.067 | .287** | .237** | −.018 | −.125** | −.407** | .169** | .031 | 1 |

Notes:

**Correlation is significant at the 0.01 level (2-tailed);

*Correlation is significant at the 0.05 level (2-tailed).

correlated with self-confidence (r = −0.163, p < 0.01). Reinforcing t-test findings that pre-elite athletes report lower self-confidence than amateur athletes. This suggests that less experienced athletes may have higher self-confidence. Comparing athletes in team versus individual sports, significant correlations were observed between sport type and both motivation (r = 0.287, p < 0.01) and team emphasis (r = 0.237, p < 0.01). These findings suggest that team sport athletes tend to have higher motivation and team emphasis than those in individual sports, reinforcing the close relationship between motivation and team orientation.

## Discussion

This study analyzed the psychometric properties of the revised Latvian version of the Psychological Skill Inventory for Sport (PSIS-R5-L), as well as the relationship between the psychological skills of different levels (elite, pre-elite and amateur) groups of athletes. In order to understand the practical value of the instrument, the PSIS-R5-L ratings of psychologists practicing in sport in relation to elite athletes were also analyzed. This approach provided an opportunity to assess the suitability of the questionnaire for practical use in the sports environment, and especially at the elite level, and also strengthened the verification of the usefulness of the instrument in the Latvian sports environment.

PSIS-R5 is one of the most commonly used tools to assess psychological skills in athletes, with the aim of identifying the most important psychological skills that allow athletes to demonstrate performance in accordance with their physical and technical abilities [11,29]. In the development of the original version, the author also points out that the PSIS-R5 measuring instrument is focused on practice, which is one of the main reasons why this measuring instrument is so relevant today. Another specific feature of the PSIS-R5 measuring instrument is that was developed with the aim of also being able to identify the psychological skills of elite-level athletes.

### Challenges and Insights in the Application of the Psychological Skills Inventory for Sports (PSIS-R5)

The previous attempt to adapt the PSIS-R5 to the Latvian sports enviroment was not fully implemented and the adapted instrument indicated several psychometric problems that required a deeper psychometric analysis [14]. The findings suggest that further, more in-depth psychometric evaluation is necessary to refine the measurement instrument. Moreover, several studies have emphasized that many instruments in sports psychology are becoming outdated, necessitating the development or revision of tools capable of accurately assessing complex psychological skills in athletes [5,6]. These observations point to the need for updated and contextually relevant instruments. To address these gaps and overcome the shortcomings of the existing version of PSIS-R5 Latvian language version, it became essential to revise and modernize the PSIS-R5. This revision aims to align the instrument with contemporary standards and ensure it effectively captures the multidimensional nature of psychological skills in sport.

One of the most significant challenges in using the PSIS-R5 is the complexity of its items content. This issue has been noted in several previous attempts to adapt the instrument. Recent studies have identified psychometric challenges related to the differentiation between psychological skills and psychological techniques. For instance, Milavic et al. developed the youth version of the PSIS (PSIS-Y) to address the specific needs of younger athletes. According to the authors, the PSIS-R5 content is not sufficiently accessible for younger athletes due to its complex language and conceptual demands. Additionally, the original version's extensive number of items has been identified as a potential limitation, reducing its practicality and effectiveness. Prior analyses of the literature have also highlighted the challenges of adapting the PSIS-R5. Various adaptations encountered difficulties in fully aligning the inventory with specific contexts, leading to the development of more suitable versions with alternative factor structures [13,11].

In this study, the adapted Latvian version of the PSIS-R5 revealed a four-factor structure. The adaptation process faced challenges, including difficulties in fully aligning the inventory with all items from the original version, resulting in some deviations from the original structure. Attempts to replicate the six-factor structure failed to yield adequate psychometric indicators or support from factor analysis, highlighting the need for a deeper examination of the inventory content.

The adapted version (PSIS-R5-L) retained four-factors from the original PSIS-R5: Self-Confidence, Motivation, Team Emphasis, and Visualization (the factor was renamed from Mental Rehearsal to Visualization based on scales content and suggestion of psychologists). The PSIS-R5-L inventory consists of 17 items in total. The Self-Confidence scale is the largest, consisting of seven items. This scale combines items from the original Anxiety Control and Concentration scales, reflecting athletes' association that these items were more closely related to self-confidence. Research supports this integration, as self-confidence has been shown to influence athletes' ability to manage anxiety and the perceived intensity of pre-competitive anxiety [30–32]. Similarly, self-confidence is closely associated with concentration abilities, which are critical in sports performance [33,34]. The remaining scales, Motivation, Team Emphasis, and Visualization, correspond to the content of the original PSIS-R5, although the number of items in each scale was reduced. This adjustment addressed the challenges identified in previous adaptations and incorporated recommendations from researchers who have used the PSIS-R5 in various sports contexts [13].

While it is notable that the adapted PSIS-R5-L does not include separate scales for Anxiety Control and Concentration, which are critical psychological skills in sports at all levels, this absence represents a gap in capturing essential constructs needed to differentiate athletes' performance effectively. However, the revised four-factor structure of the PSIS-R5-L addresses many of the challenges faced during the adaptation of the previous Latvian version. This adaptation not only builds on the strengths of the original framework but also ensures that the revised instrument is more streamlined and practical for use in sports environments. Although items related to Anxiety Control and Concentration were incorporated into the Self-Confidence factor, the omission of dedicated both scales may limit the specificity of measuring these psychological domains. Considering that both anxiety management and concentration are widely recognized as essential psychological skills for athletic success across all competitive levels, future studies should explore the possibility of supplementing the PSIS-R5-L with additional subscales or complementary measures. This would ensure a more comprehensive evaluation of athletes' psychological skill sets while preserving the practical advantages of the revised instrument.

### Investigating Psychological Skill Differences Among Athletes of Different Levels

A key question in sports psychology research has always been what distinguishes athletes of different performance levels and how they differ in psychological skills. In this context, it was important to examine whether the PSIS-R5-L could effectively identify differences in psychological skill indicators among elite, pre-elite, and amateur athletes. The findings revealed significant differences across several scales between these groups. For example, the results on the Self-Confidence scale indicated statistically significant differences between elite and pre-elite athletes, as well as between pre-elite and amateur athletes ($p < 0.05$). This suggest that perceptions of self-confidence, as captured by the scale differ significantly depending on athletes' competitive experience and achievements. This aligns with previous research highlighting the importance of context when interpreting psychological measures in sports. For example, Ita et al. found that elite athletes demonstrated significantly higher levels of confidence, visualization, anxiety control, and team cohesion compared to non-elite athletes [35]. Their study also concluded that motivation levels were equally high among elite and non-elite athletes, partially confirming the results of this study. In the current study, no statistically significant differences in motivation were found between elite and pre-elite athletes ($p > 0.05$). Significant differences were observed between pre-elite and amateur athletes, as well as between elite and amateur athletes, with amateurs showing lower levels of motivation ($p < 0.05$). Other research has offered additional insights into motivation, emphasizing that it is driven by athletes' interest and enjoyment in sports, regardless of their years of experience [36]. The results suggest that the Motivation scale distinctly varies according to the achievement levels of different athletes, highlighting its importance as a key factor when assessing athletes at varying levels of performance. This finding underscores the necessity of considering motivation as a critical dimension in psychological evaluations, as it appears to significantly differentiate athletes based on their success in sports.

Interestingly, no significant differences were observed between the groups at different performance levels on the Team Emphasis scale (p > 0.05). This lack of differentiation could be attributed to the nature of this psychological skill, which tends to be more relevant in team sports than in individual sports. Additionally, these findings may have been influenced by the composition of the elite group, which included a predominance of athletes from certain sports. Despite this, both empirical evidence and a review of the scientific literature confirm that the concept of team belonging is an important psychological construct [37,38]. As such, it should not be overlooked or excluded from comprehensive psychological assessments. Team belonging plays a critical role in fostering team dynamics, cohesion, and overall athletic performance, making it an essential dimension for evaluating athletes in team-based sports contexts.

Imagery and visualization are widely recognized as essential psychological skills in elite sports and have become a notable focus in sports psychology research. The results of this study revealed statistically significant differences in Visualization scale only between elite and amateur athletes (p < 0.05), representing athletes at the highest and lowest levels of competitive sports. Previous research on visualization within the broader context of imagery supports these findings, demonstrating that an athlete's level of competition significantly influences their ability to imagine [39,40]. Athletes with greater competitive experience tend to utilize their visualization more frequently and effectively during preparation for competitions and training [41]. This highlights the importance of visualization as a critical tool for performance enhancement, particularly for athletes at the elite level. These results suggest that the ability to mentally visualize performance is a distinct psychological skill that varies among athletes with different levels of achievement, reinforcing the importance of visualization as a critical component in psychological skill assessments.

## Psychologists' Perspectives on Elite Athletes' Psychological Skills

Sports psychologists often focus their practice on employing instruments to measure sports-related behaviors, particularly psychological skills and strategies. These tools have gained considerable attention due to their ability to distinguish between more and less successful athletes while also providing valuable evidence for the effectiveness of psychological skills training programs. Among these instruments, the PSIS-R5 stands out as a key tool for assessing critical psychological attributes in athletes. The significance of such assessments extends beyond athletes' self-perception, it also includes evaluations from sports psychologists, offering a deeper and more nuanced understanding of the dimensions measured by instruments like the PSIS-R5. This perspective is especially relevant in professional and high-performance sports, where psychological skills are often the main factor to competitive success. Several researchers have highlighted the complexity of the inventory's item construction, underscoring the need for further psychometric evaluation [13,42]. Therefore, it is essential to develop a tool that accurately measures psychological skills in sports contexts. Such an instrument would enable precise assessment of psychological skill levels within the sports population and identify meaningful differences across varying levels of athletic groups. Recent developments in sport psychology emphasize psychological skills such as emotional regulation, resilience, and dynamic concentration as critical components of athletic performance in contemporary sports environments [43]. Several excluded items reflected narrower or less dynamic concepts that may no longer align with current understandings of psychological skill development. Therefore, the PSIS-R5-L retains the core strengths of the original inventory while optimizing its relevance to modern mental skills training and assessment needs.

In this study, a comparative analysis of PSIS-R5-L data and the ratings provided by Latvian psychologists revealed intriguing insights. In most psychological skill assessments, the responses of elite athletes and psychologists were similar. However, the Self-Confidence scale emerged as an exception. Psychologists observed significantly lower levels of self-confidence among elite athletes compared to athletes' self-ratings on the PSIS-R5-L. This discrepancy raises critical questions about the subjective nature of self-ratings and possible biases that athletes may have when assessing their psychological skills. Athletes may rate themselves high due to performance results or ingrained perceptions of their abilities. In contrast, psychologists often use a more holistic and nuanced approach, identifying insecurities and pressures in athletes. It must be admitted that such an approach has been very little examined and studied in research. Jekauc et

al. indicate that elite athletes, despite their achievements, often experience hidden insecurities or stressors, especially in environments with high demands [44]. This highlights the limitations of conducting psychological assessments relying solely on self-reported data. Combining self-reports with professional assessments provides a more comprehensive and accurate profile of an athlete's psychological skills, improving the utility of tools such as the PSIS-R5-L in both research and practical settings.

While the PSIS-R5-L provides a valuable framework for psychological skill assessment, its practical implementation in real-world sports contexts presents unique challenges. Psychologists working with athletes must consider not only the athlete's skill level but also the specific demands of their sport. For example, individual-sport athletes may require more focused interventions on self-regulation and visualization, while team-sport athletes may benefit from enhanced strategies for communication and team emphasis. Athletes at different competitive levels often differ in their access to psychological support and openness to self-reflection, all of which can influence the interpretation and use of assessment results. In the Latvian context, where access to regular sport psychology services remains limited. Practitioners must often tailor feedback and interventions using brief and resource-efficient tools like the PSIS-R5-L. This makes the tool especially valuable for psychologists needing a structured and concise method to inform psychological skills training, while also highlighting the importance of adapting interpretations based on sport type, athlete maturity, and performance level.

## Limitation of the study

The development of the PSIS-R5-L followed survey adaptation guidelines, as well as the original instrument's development guidance. While the adaptation adhered to a rigorous procedure and the study sample is notably large for the Latvian athlete population, it is essential to recognize potential limitations that may affect the study's outcomes.

One limitation of survey-based research is the reliance on self-reported data, which may introduce inaccuracies, thereby limiting the objectivity of the provided responses of athletes. Respondents might overestimate or underestimate their psychological skills, influenced by personal perceptions or situational factors.

Another limitation involves the relatively small sample of elite athletes (n = 25) included in the study. Despite the overall sample size being sufficient, expanding the elite athlete group would allow for a more detailed exploration of significant psychological skill characteristics unique to this population. Future research should prioritize increasing the number of participants in this group to enhance the reliability of findings and uncover additional insights into elite athletes' psychological profiles. Additionally, the limited sample size of psychologists (n = 5) presents another constraint. While their professional evaluations provide valuable perspectives, a larger sample of psychologists could offer a more comprehensive understanding of elite athletes' psychological skills from the practitioners' viewpoint. However, the scarcity of such specialists within the Latvian context presents a challenge. Encouraging collaboration with psychologists in other regions or international experts might address this limitation.

Given these considerations, it is imperative to encourage further research on the PSIS-R5-L, focusing particularly on elite athletes and integrating broader practitioner perspectives. Future studies should explore the issues identified in this research, such as the discrepancies between self-assessment and external evaluations, while continuing to refine and validate the instrument for diverse sports contexts.

## Conclusion

This study focused on developing and adapting the internationally recognized Psychological Skills Inventory for Sport (PSIS-R5) for the Latvian sports environment, resulting in the PSIS-R5-L. The adapted version reliably and validly assesses the psychological skills of athletes across different levels of athletic achievement, including elite, pre-elite, and amateur athletes. The PSIS-R5-L comprises 17 items grouped into four scales: Self-Confidence, Motivation, Team Emphasis, and Visualization. Its factor structure aligns with a modern and contextually relevant approach to understanding psychological skills within the Latvian sports environment.

The psychometric properties of the PSIS-R5-L confirms the suitability of the adapted measurement instrument for practical use. Importantly, the instrument addresses the unique characteristics of modern athletes and provides valuable insight into their psychological skills at different levels of performance. Its application offers new insights into athlete evaluation, highlights significant differences in achievement levels, and is further validated by assessments from sports psychologists. These results reinforce the instrument's relevance and effectiveness in professional practice. Although the adaptation was conducted in Latvia, the PSIS-R5-L contributes to the broader field of sport psychology and offers a foundation for future cross-cultural applications and validations. While the adaptation of the PSIS-R5-L was based on a large and diverse athlete sample (n = 444), future studies could further explore its application within specific subgroups, such as elite athletes, to strengthen group-level interpretations. Additionally, given the varying psychological demands across different types of sports, further validation across distinct sporting contexts is recommended.

## Practical Implementation

The practical contribution of this study is evident in its potential to enhance the work of sports psychologists, psychological preparation coaches, and sports specialists who aim to promote psychological preparation of athletes. Self-reported measurement instruments hold significant value for sports practice. These tools can provide accurate and in-depth insights into athletes' psychological states, enabling practitioners to identify specific psychological characteristics, including strengths and potential barriers that may impact performance. Valid and reliable measurement instruments, such as the PSIS-R5-L, are essential for uncovering nuanced psychological constructs, offering practitioners a deeper understanding of athletes' psychological skills and mental state.

Another critical practical contribution of this study lies in its emphasis on psychological skills training. The results indicate that scores on various scales are not particularly high or close to maximum levels, highlighting the necessity of integrating targeted psychological skills training into athletes' development programs. Tailoring such training to suit the athletes' competitive level can enhance their psychological skills and overall performance, underscoring the importance of psychological preparation as a core component of athletic success.

Finally, the study provides a significant contribution to the field of sports science by adapting a scientifically validated psychological skills measurement instrument to the Latvian sports context. The results obtained offer evidence-based insights that can guide future psychological assessments and training methodologies, fostering a deeper understanding of the psychological dimensions of athletic performance in Latvia. Given the instrument's psychometric performance and the widespread use of the PSIS-R5 framework globally, the PSIS-R5-L may also serve as a reference point for future adaptations in other cultural contexts, thereby supporting the development of universally applicable psychological assessment tools in sport.

## Supporting information

**S1 File. Latvian adapted version of Psychological Skills Inventory for Sport (PSIS-R5-L) and the original version of PSIS-R5.**
(PDF)

**S2 File. List of excluded items from the original PSIS-R5 inventory.**
(PDF)

## Acknowledgments

We sincerely thank the members of the Latvian Sports Psychology Association for their participation in the study and their valuable feedback.

## Author contributions

**Conceptualization:** Katrina Volgemute, Gundega Ulme, Viktorija Perepjolkina, Agita Abele.

**Data curation:** Katrina Volgemute, Gundega Ulme, Renars Licis, Agita Abele, Rodrigo Lavins, Anna Tihija, Everts Grants.

**Formal analysis:** Katrina Volgemute, Gundega Ulme, Viktorija Perepjolkina.

**Funding acquisition:** Katrina Volgemute.

**Investigation:** Katrina Volgemute.

**Methodology:** Katrina Volgemute, Gundega Ulme.

**Software:** Katrina Volgemute, Viktorija Perepjolkina.

**Supervision:** Katrina Volgemute.

**Validation:** Katrina Volgemute, Gundega Ulme, Viktorija Perepjolkina, Renars Licis.

**Visualization:** Katrina Volgemute.

**Writing – original draft:** Katrina Volgemute, Gundega Ulme, Viktorija Perepjolkina, Renars Licis, Agita Abele, Rodrigo Lavins.

**Writing – review & editing:** Katrina Volgemute, Gundega Ulme, Viktorija Perepjolkina, Renars Licis, Agita Abele, Rodrigo Lavins, Alīna Klonova.

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
