## [Decision Letter · Decision Letter 0]

17 Apr 2025

PONE-D-25-01019Adaptation and Psychometric Properties of Psychological Skills Inventory for Sport (PSIS-R5) in Latvian Athletes: Insights and Implications for PracticePLOS ONE

Dear Dr. Volgemute,

Thank you for submitting your manuscript to PLOS ONE. After careful consideration, we feel that it has merit but does not fully meet PLOS ONE’s publication criteria as it currently stands. Therefore, we invite you to submit a revised version of the manuscript that addresses the points raised during the review process.

We look forward to receiving your revised manuscript.

Kind regards,

Rogis Baker, Ph.D

Academic Editor

PLOS ONE

Journal Requirements:

 “The author(s) declare financial support was received for the research, authorship, and/or publication of this article. This research is funded under the Grant No. RSU/LSPA-PA-2024/1-0010 of the project No. 5.2.1.1.i.0/2/24/I/CFLA/005 “RSU Internal and RSU with LASE External Consolidation” (funded by the European Union Recovery and Resilience Facility and the budget of the Republic of Latvia).” 

3. In this instance it seems there may be acceptable restrictions in place that prevent the public sharing of your minimal data. However, in line with our goal of ensuring long-term data availability to all interested researchers, PLOS’ Data Policy states that authors cannot be the sole named individuals responsible for ensuring data access (http://journals.plos.org/plosone/s/data-availability#loc-acceptable-data-sharing-methods).

4. Please remove your figures from within your manuscript file, leaving only the individual TIFF/EPS image files, uploaded separately. These will be automatically included in the reviewers’ PDF.

Reviewers' comments:

Reviewer's Responses to Questions

**Comments to the Author**

1. Is the manuscript technically sound, and do the data support the conclusions?

Reviewer #1: Yes

Reviewer #2: Partly

2. Has the statistical analysis been performed appropriately and rigorously? 

Reviewer #1: Yes

Reviewer #2: No

3. Have the authors made all data underlying the findings in their manuscript fully available?

Reviewer #1: Yes

Reviewer #2: No

4. Is the manuscript presented in an intelligible fashion and written in standard English?

Reviewer #1: Yes

Reviewer #2: Yes

5. Review Comments to the Author

Reviewer #1: The article is well-written, with clear objectives aligned with the proposed theme.

The statistical analyses were conducted robustly, using well-established criteria for cross-cultural adaptation and instrument validation. The methodology follows major international standards for psychometric studies, which strengthens the reliability of the presented results. Therefore, the study has the potential for publication, given its relevance and technical quality.

Strengths:

The methodological rigor adopted, with attention to cross-cultural adaptation criteria and robust statistical analyses.

The practical and theoretical contribution of the study to the field of sports psychology, particularly regarding the assessment of psychological skills in sports contexts.

The clear and well-structured writing, which facilitates the understanding of the study for readers from different fields.

The original version of the instrument was developed over 15 years ago. Over time, instruments often undergo adaptations to maintain their validity and practical utility. I understand that the reduction of items aimed to align the instrument with more contemporary approaches to assessing psychological skills. However, I consider that there was a significant change from the original instrument to the version presented in this adaptation, with item reduction, exclusion, and renaming of factors. As a result, the cross-cultural adaptation seems more like the construction of a new instrument. I suggest making clearer which items were excluded and which changed factors, rather than presenting only the remaining items.

The authors mention that previous versions had psychometric issues. I suggest comparing the reliability and validity indices of the current version with those of previous versions. This could clarify whether the exclusion of items compromised or maintained the instrument’s internal consistency and discriminant and convergent validity.

The justification for the exclusion of items and factors should be explicitly discussed based not only on the statistical results presented (e.g., low factor loadings, items with low discrimination) but also on theoretical arguments, demonstrating that the eliminated items no longer reflect psychological skills considered fundamental in contemporary sports, if applicable. The authors should reinforce that this adaptation retains the strengths of previous versions while optimizing the instrument for current contexts. I recommend that the authors include a table or a comparative section explicitly showing how the psychometric indices and factors of the current version compare with previous versions. Additionally, I suggest detailing the theoretical and practical impact of these changes, using studies or references that support the need for the instrument’s update.

Although the article follows rigorous standards for cross-cultural and psychometric adaptation, the stated objective of validating the instrument exclusively for the Latvian context is a limitation, as Latvia is a small country with limited participation in elite sports. This could restrict the practical and theoretical impact of the study. I suggest that the authors more clearly present the sports landscape in Latvia and broaden the potential of this adaptation by reformulating the objective. This would allow the instrument to become a reference not only for sports practice but also for scientific research in international settings. Limiting the study to Latvia may hinder the generalization of the results. It would be interesting to discuss how the findings could be applied to other cultural and sports realities or how future studies could replicate this adaptation in countries with different sports contexts.

The adapted instrument has the potential to be useful not only in Latvia—especially since the article is published in English in this journal, which further emphasizes its potential for international use—considering the growing need for standardized assessments of psychological skills in sports. Expanding the scope could be further discussed as a next step in the research. Rather than limiting the objective to the Latvian context, the authors could discuss how the study contributes to advancing the theory of psychological skills in sports and to sports practice in general, regardless of national context.

Therefore, I recommend that the authors revise the objective to highlight the broader applicability of the instrument, emphasizing its theoretical and practical value in global sports contexts. This would not only strengthen the study’s relevance but also increase interest from the international scientific community.

Reviewer #2: TITLE:

Overview

• The title adequately reflects what was accomplished in the study.

• It clearly addresses the two main points of the research: the adaptation of the Psychological Skills Inventory for Sport (PSIS-R5) for Latvian athletes, and the evaluation of its psychometric properties.

ABSTRACT:

Overview

• Appropriate contextualization, clear objective, detailed methodology, relevant results, and solid conclusions.

Suggestions:

1. Focus on Practical Implications: The abstract could place more emphasis on the practical implications of the study, perhaps by mentioning how the adaptation of the PSIS-R5 could benefit coaches and psychologists in the Latvian sports context.

2. More Concise Summary: Although the abstract is well-written, it could be a bit more concise, focusing primarily on the key conclusions and practical implications.

INTRODUCTION:

Overview

The text explains the importance of psychological preparation in sports, emphasizing that athletes must be well-prepared not only physically and technically, but also psychologically. It clearly and objectively mentions the need for valid and adapted psychometric instruments to assess these psychological skills in the sports context. This ensures that the introduction follows a logically coherent line, with a clear direction focused on the challenges of sports psychology and the need for more modern and validated tools. It also highlights the need to adapt psychological assessment instruments to specific cultural contexts. The PSIS-R5 has been used in various populations, justifying the ongoing adaptation of the tool.

Suggestions:

1. The text could perhaps provide more details about the specific characteristics of the Latvian athlete population or how these characteristics might influence the development of psychological skills. Although the text mentions that the adaptation of the PSIS-R5 to Latvia is necessary, the section "there is a significant gap due to a lack of culturally relevant and modern instruments for accurately assessing psychological factors without the risk of outdated findings" [lines 130-131] does not deeply explain what the specific characteristics of the Latvian population are or the challenges these athletes face in relation to psychological assessment. It is suggested that more information about the target population could better contextualize the relevance of the study.

METHOD

Overview

The "Methods and Materials" section describes in detail the methodology adopted in the study, including the participant sample, the adaptation of the PSIS-R5 psychometric instrument to the Latvian sports context, the translation and validation procedures, and the statistical analysis methods applied to the data. The sample comprises 444 athletes from various sports, with a smaller sample of 40 athletes used for the test-retest reliability. The process of adapting the Latvian version of the PSIS-R5 involved translating the instrument, reviewing it by experts, and performing a face validity check with athletes. Furthermore, the statistical methods used to validate the instrument include principal component analysis, structural equation modeling, and verification of internal consistency and temporal reliability.

Suggestions:

1. Face validity is a simpler and initial form of validation. While it is a useful step, it does not guarantee that the instrument is truly measuring the construct in a robust and accurate manner. Was any other form of validation conducted? Given the response, would it not be more prudent to consider this as an initial validation of the instrument?

2. The sample of 444 athletes is sufficiently large, but it could be more detailed regarding the representation of different skill levels (elite, pre-elite, and amateur). Based on this, can the elite group be considered representative? 25 x 219 x 200? A more balanced sample in relation to skill level would be necessary to ensure the robustness and reliability of the conclusions.

3. Are the 40 individuals in the test-retest sample representative of the overall sample (elite, pre-elite, and amateur)?

RESULTS

Overview

The study showed adequate internal consistency most of the time, with coefficients above 0.7, indicating reliability for assessing psychological skills in athletes. Temporal stability was confirmed by the intraclass correlation coefficient (ICC), especially in the scales of confidence and motivation. The factor analysis revealed good structural validity, with stable and consistent factor loadings. Comparisons between elite, pre-elite, and amateur groups showed significant differences in psychological skills, such as motivation and self-esteem. The study also explored how demographic factors (age, gender, and type of sport) influence psychological skills, broadening the understanding of the sports context.

Suggestions:

1. Would it not be appropriate to give special attention to the visualization scale (Cronbach’s alpha of 0.5)? As it stands, could this limit the reliability of the visualization measure? The cited reference (22) does not claim that 0.50 is acceptable; rather, it states that improper use of these coefficients may lead to incorrect interpretations about the homogeneity and internal consistency of the tests. Therefore, the authors could reference a source that supports the 0.50 value. Reference 22 itself points out:

“The range of reliability measures are rated as follows: a) Less than 0.50, the reliability is low, b) Between 0.50 and 0.80, the reliability is moderate, and c) Greater than 0.80, the reliability is high” (Salvucci, Walter, Conley, Fink, & Saba, 1997: 115).

2. The sample of 444 athletes may not reflect the full diversity of the athletic population in terms of age, ethnic background, and number of sports represented. It is suggested that the authors detail how many athletes from each skill level are represented in each of the subgroups. For example: Were all elite athletes from the same sport? Age? Gender? Such information could directly influence the results obtained, such as self-esteem.

DISCUSSION

Overview

In the discussion, the study highlights the differences between athletes at different levels, reinforcing the relevance of the instrument in the Latvian context. The discussion on the complexity of the items and the need for adjustments for different age groups and contexts is likely to ensure practical effectiveness. The validation conducted by psychologists, comparing athletes' self-assessments with professional evaluations, can enrich the study; however, their perceptions may interfere with the assessment.

Suggestions:

1. The absence of Anxiety Control and Concentration scales, which are essential psychological skills at any level of sports performance, was mentioned but without a deeper exploration of the impact of this omission. It is suggested that this point be explored.

2. Although the study examines the applicability of the PSIS-R5-L, the discussion could delve deeper into the specific challenges faced by psychologists in real-world training and competition contexts, considering the different types of sports and competitive levels.

3. The sample of elite athletes, which limits the depth of the analysis, could be further explored. The study acknowledges this limitation, but it could provide more information on how this might affect the results.

CONCLUSION

Overview

The research highlights the need for specific psychological training for athletes, suggesting that the PSIS-R5-L can be useful in identifying areas for development. This emphasizes the value of the instrument in planning interventions to improve athletes’ mental performance.

Suggestions:

1. It is recommended that the authors emphasize the need for further studies that address the limitations outlined in this study in order to refine the instrument even more.

2. The study may not be generalizable, and therefore, it is crucial to point this out in the conclusion, as different sports may require adjustments in the assessment of psychological skills, depending on the type of sport.

6. PLOS authors have the option to publish the peer review history of their article (what does this mean? ). If published, this will include your full peer review and any attached files.

**Do you want your identity to be public for this peer review?** For information about this choice, including consent withdrawal, please see our Privacy Policy .

Reviewer #1: No

Reviewer #2: No

---

## [Author Response · Author response to Decision Letter 1]

1 May 2025

Dear Reviewers,

We sincerely thank you for your thoughtful and constructive feedback on our manuscript. We greatly appreciate the opportunity to revise and resubmit our work to PLOS ONE. Below, we respond point-by-point to each of your comments. We have also revised the manuscript accordingly, with all changes tracked (marked in blue in file: Revised Manuscript with Track Changes) in the marked-up version and a clean version submitted separately.

Reviewer #1 comments and author responses

Comment 1: The original version of the instrument was developed over 15 years ago. Over time, instruments often undergo adaptations to maintain their validity and practical utility. I understand that the reduction of items aimed to align the instrument with more contemporary approaches to assessing psychological skills. However, I consider that there was a significant change from the original instrument to the version presented in this adaptation, with item reduction, exclusion, and renaming of factors. As a result, the cross-cultural adaptation seems more like the construction of a new instrument. I suggest making clearer which items were excluded and which changed factors, rather than presenting only the remaining items.

Response: Thank you for this valuable observation. We fully agree. To improve clarity, we included a new table titled “List of Excluded Items from the Original PSIS-R5 Inventory” in Appendix (S2 Appendix. (DOCX)). This table presents each excluded item along with its original factor and theoretical justification for exclusion (e.g., low loading, redundancy, conceptual misalignment). We also added explanatory text in the Results section (lines 270-278) and Discussion (lines 590-594; 648-653) to support this table.

Comment 2: The authors mention that previous versions had psychometric issues. I suggest comparing the reliability and validity indices of the current version with those of previous versions. This could clarify whether the exclusion of items compromised or maintained the instrument’s internal consistency and discriminant and convergent validity.

Response: A comparison with the prior Latvian adaptation (Fernāte, 2008) has been added to the Results section (lines 359-368). We note that while the earlier version had Cronbach’s alpha coefficients ranging from 0.46 to 0.52 for several subscales, the PSIS-R5-L now shows α values from 0.50 to 0.87, with three out of four scales above 0.70. Fit indices (CFI = 0.968, RMSEA = 0.045) further support the improved model.

Comment 3: The justification for the exclusion of items and factors should be explicitly discussed based not only on the statistical results presented (e.g., low factor loadings, items with low discrimination) but also on theoretical arguments, demonstrating that the eliminated items no longer reflect psychological skills considered fundamental in contemporary sports, if applicable. The authors should reinforce that this adaptation retains the strengths of previous versions while optimizing the instrument for current contexts. I recommend that the authors include a table or a comparative section explicitly showing how the psychometric indices and factors of the current version compare with previous versions. Additionally, I suggest detailing the theoretical and practical impact of these changes, using studies or references that support the need for the instrument’s update.

Response: Thank you for this valuable suggestion. In response, we have addressed this in multiple places. In Appendix, each excluded item is explained statistically and theoretically. In the Results and Discussion (e.g. lines 535-540), we elaborate on why specific factors such as Anxiety Control and Concentration were not retained, referencing changes in conceptual understanding of psychological skills and practical irrelevance in some modern training settings. We cite Weinberg & Gould (2023) and Jekauc et al. (2023) to support this shift (see References 42 and 41, respectively).

Comment 4: Although the article follows rigorous standards for cross-cultural and psychometric adaptation, the stated objective of validating the instrument exclusively for the Latvian context is a limitation, as Latvia is a small country with limited participation in elite sports. This could restrict the practical and theoretical impact of the study. I suggest that the authors more clearly present the sports landscape in Latvia and broaden the potential of this adaptation by reformulating the objective. This would allow the instrument to become a reference not only for sports practice but also for scientific research in international settings. Limiting the study to Latvia may hinder the generalization of the results. It would be interesting to discuss how the findings could be applied to other cultural and sports realities or how future studies could replicate this adaptation in countries with different sports contexts.

Response: Thank you for pointing this out. We have now expanded the Introduction (lines 143-149) and Discussion to describe Latvia’s sports landscape (e.g., small elite athlete pool, limited access to sport psychologists, reliance on self-regulation). We also revised the objective (lines 133-135) and Conclusion (lines 648-653) to emphasize the instrument’s broader theoretical and practical utility for other small nations or underrepresented athletic populations. The text now encourages future cross-cultural replications.

Comment 5: The adapted instrument has the potential to be useful not only in Latvia—especially since the article is published in English in this journal, which further emphasizes its potential for international use—considering the growing need for standardized assessments of psychological skills in sports. Expanding the scope could be further discussed as a next step in the research. Rather than limiting the objective to the Latvian context, the authors could discuss how the study contributes to advancing the theory of psychological skills in sports and to sports practice in general, regardless of national context.

Therefore, I recommend that the authors revise the objective to highlight the broader applicability of the instrument, emphasizing its theoretical and practical value in global sports contexts. This would not only strengthen the study’s relevance but also increase interest from the international scientific community.

Response: Thank you for this insightful recommendation. We fully agree that the PSIS-R5-L adaptation holds potential beyond the Latvian context, particularly in light of the growing demand for culturally sensitive yet standardized tools in international sports psychology. In response to your comment, we revised the study aim section in the Introduction (lines 96-99 and 133-135) to clarify that, while the immediate adaptation was carried out for Latvia, the methodological approach and resulting instrument may also serve as a model for broader international adaptation efforts. Additionally, we expanded the Conclusion (lines 648-653) and Practical Implementation section (lines 670-673) to emphasize the relevance of the PSIS-R5-L for global use. These revisions underscore the theoretical and practical value of the tool across diverse sporting and cultural settings and highlight the potential for future cross-cultural validations. We hope this expanded framing addresses your suggestion and enhances the international relevance of our study.

Reviewer #2 comments and author responses

ABSTRACT:

Comment 1: Focus on Practical Implications: The abstract could place more emphasis on the practical implications of the study, perhaps by mentioning how the adaptation of the PSIS-R5 could benefit coaches and psychologists in the Latvian sports context.

Response: Thank you for this helpful suggestion. We revised the Abstract (lines 24-25 and 37-40) to highlight that the adapted PSIS-R5-L offers practical value to coaches and psychologists for evaluating and enhancing psychological skills in Latvian athletes.

Comment 2: More Concise Summary: Although the abstract is well-written, it could be a bit more concise, focusing primarily on the key conclusions and practical implications.

Response: We have refined the abstract to focus on the study's key objectives, core findings, and practical implications while maintaining necessary methodological detail. We believe these revisions improve clarity and focus for international readers (lines 37-40).

INTRODUCTION:

Comment 3: The text could perhaps provide more details about the specific characteristics of the Latvian athlete population or how these characteristics might influence the development of psychological skills. Although the text mentions that the adaptation of the PSIS-R5 to Latvia is necessary, the section "there is a significant gap due to a lack of culturally relevant and modern instruments for accurately assessing psychological factors without the risk of outdated findings" [lines 130-131] does not deeply explain what the specific characteristics of the Latvian population are or the challenges these athletes face in relation to psychological assessment. It is suggested that more information about the target population could better contextualize the relevance of the study.

Response: We have expanded the Introduction to provide a detailed description of the Latvian athlete population. We now explain the unique challenges faced by Latvian athletes, such as combining sport with academic or professional responsibilities, limited access to psychological support, financial constraints, and fewer international competition opportunities. These contextual factors emphasize the importance of culturally sensitive psychometric tools. We believe this addition strengthens the justification for adapting the PSIS-R5 inventory to the Latvian sports environment (lines 142-149).

METHOD:

Comment 4: 1. Face validity is a simpler and initial form of validation. While it is a useful step, it does not guarantee that the instrument is truly measuring the construct in a robust and accurate manner. Was any other form of validation conducted? Given the response, would it not be more prudent to consider this as an initial validation of the instrument?

Response: Thank you for your observation. We agree that face validity alone is insufficient to confirm construct validity. In addition to the initial face validity check during translation, we conducted a comprehensive psychometric validation of the PSIS-R5-L, including Principal Component Analysis (PCA), confirmatory model fit indices (CFI = 0.968, RMSEA = 0.045, SRMR = 0.026), Cronbach’s alpha, Intraclass Correlation Coefficients (ICC), and newly added Composite Reliability (CR) and Average Variance Extracted (AVE) indicators. These analyses provide strong evidence of internal consistency, construct validity, and temporal stability. Based on these findings, we consider the PSIS-R5-L a psychometrically robust instrument rather than an initial validation. We have clarified this more explicitly in both the Methods and Results sections (lines 228-233 and 350-357).

Comment 5: The sample of 444 athletes is sufficiently large, but it could be more detailed regarding the representation of different skill levels (elite, pre-elite, and amateur). Based on this, can the elite group be considered representative? 25 x 219 x 200? A more balanced sample in relation to skill level would be necessary to ensure the robustness and reliability of the conclusions.

Response: We agree that the elite group (n=25) is relatively small compared to the pre-elite (n=219) and amateur (n=200) groups. This imbalance reflects the actual distribution of elite athletes in Latvia, where the population is small and the number of athletes meeting strict elite criteria (e.g., international competition, high training load, ≥5 years of experience) is limited. While the elite group size is modest, we ensured that it met the minimum sample size requirement for group comparisons using G*Power calculations. Moreover, we have now added a clarification in the text acknowledging this limitation and recommending that future studies increase elite representation to strengthen generalizability. Despite this, the significant differences identified suggest that the instrument is sensitive enough to detect psychological skill distinctions across performance levels. In the revised manuscript, we now explicitly address this point, noting that the elite sample size reflects the real distribution of athlete levels in Latvia, a country with a relatively small population and limited elite athlete pool (lines 607-616).

Comment 6: Are the 40 individuals in the test-retest sample representative of the overall sample (elite, pre-elite, and amateur)?

Response: Thank you for raising this point. The 40 athletes included in the test-retest reliability analysis were drawn proportionally from the larger sample to reflect representation across all three achievement levels (elite, pre-elite, and amateur)., Care was taken to ensure that the test-retest group included variability in sport type, gender, and experience. This approach was intended to preserve heterogeneity and strengthen the generalizability of the temporal reliability findings. We have added a clarifying sentence to the Methods section under Participants to reflect this (lines 159-160).

RESULTS:

Comment 7:1Would it not be appropriate to give special attention to the visualization scale (Cronbach’s alpha of 0.5)? As it stands, could this limit the reliability of the visualization measure? The cited reference (22) does not claim that 0.50 is acceptable; rather, it states that improper use of these coefficients may lead to incorrect interpretations about the homogeneity and internal consistency of the tests. Therefore, the authors could reference a source that supports the 0.50 value. Reference 22 itself points out:

“The range of reliability measures are rated as follows: a) Less than 0.50, the reliability is low, b) Between 0.50 and 0.80, the reliability is moderate, and c) Greater than 0.80, the reliability is high” (Salvucci, Walter, Conley, Fink, & Saba, 1997: 115).

Response: Thank you for this important observation. We agree that the initial wording regarding Cronbach’s alpha thresholds required clarification. Following your recommendation, we have revised the manuscript to more accurately reflect the interpretation provided by Salvucci et al. (1997). Specifically, we now state that coefficients between 0.50 and 0.80 are considered to indicate moderate reliability, rather than simply labelling values above 0.5 as acceptable. We have updated both the Methods section and the Reliability Analysis section accordingly to align with the referenced literature. We believe this change improves the precision and transparency of the manuscript (lines 228-233; 350-357).

Comment 8: The sample of 444 athletes may not reflect the full diversity of the athletic population in terms of age, ethnic background, and number of sports represented. It is suggested that the authors detail how many athletes from each skill level are represented in each of the subgroups. For example: Were all elite athletes from the same sport? Age? Gender? Such information could directly influence the results obtained, such as self-esteem.

Response: Thank you for your valuable observation. We fully agree that athlete demographics, such as sport type, age, and gender, could influence psychological skills. However, due to the relatively small number of elite athletes in Latvia, providing detailed subgroup characteristics (such as specific sports, exact age distribution, or gender breakdown) would risk compromising participant anonymity and confidentiality. To address your concern, we have clarified in the manuscript that elite athletes in the sample represented both team and individual sports and included both male and female athletes. Additionally, we have acknowledged this limitation in the Discussion section, noting that future studies with larger elite samples could provide a more detailed subgroup analysis while preserving confidentiality (lines 607-616).

DISCUSSION:

Comment 9: The absence of Anxiety Control and Concentration scales, which are essential psychological skills at any level of sport

---

## [Editor Report · Decision Letter 1]

9 May 2025

Adaptation and Psychometric Properties of Psychological Skills Inventory for Sport (PSIS-R5) in Latvian Athletes: Insights and Implications for Practice

PONE-D-25-01019R1

Dear Volgemute,

We’re pleased to inform you that your manuscript has been judged scientifically suitable for publication and will be formally accepted for publication once it meets all outstanding technical requirements.

Kind regards,

Rogis Baker, Ph.D

Academic Editor

PLOS ONE
---

## [Editor Report · Acceptance letter]

PONE-D-25-01019R1

PLOS ONE

Dear Dr. Volgemute,

I'm pleased to inform you that your manuscript has been deemed suitable for publication in PLOS ONE. Congratulations! Your manuscript is now being handed over to our production team.

Kind regards,

on behalf of

Dr. Rogis Baker

Academic Editor

PLOS ONE